# Analysis of the spatial and temporal evolution process and development trend of innovation capability of Chinese sports industry

**Jiang Chen**, **ChuanMing Cai** *, **TingTing Zhang**

School of Physical Education, JiMei University, Xiamen, Fujian, China

* cj18159506337@163.com

**Data Availability Statement:** The data are held in a public repository. URL:https://psssystem.cponline.cnipa.gov.cn/conventionalSearch. All patents files are available from the Patent Search and Analysis

## Abstract

The paper uses spatial panel data on the number of patents in the field of sports in each province from 2017–2022 to explore the spatial and temporal evolution process of the innovation capacity of China's sports industry and analyze its development trend based on theories related to economic geography, using exploratory spatial data analysis (ESDA) and Spatial-temporal correlation analysis research methods. The study shows that: 1. there is a large spatial regional disparity in the innovation capacity of China's sports industry, showing an obvious pattern of strength in the east and weakness in the west. 2. there is an obvious phenomenon of spatial agglomeration in the innovation capacity of China's sports industry and the agglomeration phenomenon in space is gradually increasing. 3. there is a spatial spillover effect in the innovation capacity of China's sports industry, and the inward spillover continues to strengthen mainly into Zhejiang, Fujian and Jiangsu provinces, while the outward spillover mainly goes to Guangdong, Hubei, Hebei, Hunan, Sichuan. Sports industry innovation capacity development put forward three suggestions: to promote the development of sports industry in the northwest region; to play the advantages of regional characteristics of sports industry; to accelerate the integration of neighboring regions to build the process. in order to help the high-quality development of sports industry.

## 1. Introduction

Innovation is the first driving force leading China's economic growth and an important connotation of high-quality development, and it is one of the important grasps to enhance the innovation capacity of cities to achieve the 14th Five-Year Plan and Long-Range Objectives Through the Year 2035 [1]. As China's economy enters a new era of quality first, the economy has shifted from a high growth stage to a high-quality development stage, and China is still in the process of "shifting" to a high-quality development stage, which is not really a high-quality development stage yet [2]. The country also attaches increasing importance to the improvement of regional independent innovation capacity, and strategic emerging industries with innovation as the main driving force are an important driving force and new pillar for economic and social development and industrial transformation and upgrading [3]. General

System database. The data in this manuscript were obtained from Patent Search and Analysis System which built by the State Intellectual Property Office of China. The relevant data are open data, and the use of them is not restricted in any way. Anyone be able to collect data on website.

**Funding:** The author(s) received no specific funding for this work.

**Competing interests:** The authors have declared that no competing interests exist.

Office of The State Council, PRC issued the "Outline of the National Strategy of Innovation-driven Development", pointing out that innovation-driven means that innovation becomes the first driving force leading development, combining scientific and technological innovation with institutional innovation, management innovation, business model innovation and cultural innovation, promoting the change of development mode to rely on continuous knowledge accumulation, technological progress and labor quality improvement, and promoting the evolution of the economy to a more advanced form, finer division of labor and more rational structure [4].

According to statistics, the total scale (total output) of the national sports industry in 2021 is 311.75 billion yuan, and the value added is 1,224.5 billion yuan. Compared with 2020, the total output of the sports industry increased by 13.9% and the added value increased by 14.1% [5]. The science and technology revolution led by the innovation-driven strategy will provide stronger scientific and technological support for sports development. Although the high-quality development of sports industry has achieved certain results, but the unbalanced and insufficient development of domestic sports is still prominent, the reform task of key areas and key links is still arduous, and the innovation capacity of sports is not yet adapted to the requirements of high-quality development [6]. Therefore, innovation-driven is the general trend of high-quality development of the sports industry. The traditional development momentum is weakening, and the crude growth mode is unsustainable [7]. DJI is a very successful case. DJI pays great attention to innovation capability [8]. DJI has accumulated more than 4,600 patent applications. In terms of international patent applications, DJI ranks 29th in the world. DJI accounted for over 80% of the global market share in 2020 and over 70% domestically, ranking first among global civil drone companies [9]. In the four years from 2013 to 2017, DJI's revenue has grown from 820 million all the way to 17.57 billion, and moreover, in 2020, it achieved a revenue of 26 billion [10]. The sports industry will be a pillar industry for the development of the national economy, and must rely on innovation-driven to create a new engine for development, continuously improve the quality and efficiency of the development of China's sports industry, open up a new space for the development of the sports industry, and achieve high-quality development of the sports industry.

Patent is an important indicator that reflect technological innovation activities, which is the most direct and main manifestation of innovation realization [11, 12]. Patents are an important manifestation of innovation output and it widely used to assess national, regional or innovation levels and innovation competitiveness [13]. Therefore, this paper analyzes the spatial evolution and the development trend of China sports industry which based on the spatial panel data of the number of patents in the sports domain in China from 2017 to 2021. Finally, based on the above research, we hope to provide ideas for the high-quality development of sports industry.

## 2. Data sources and research methods

### 2.1 Data sources

The data in this paper were obtained from Patent Search and Analysis System which built by the State Intellectual Property Office of China. Obtain data on patents valid in the field of sports from January 1, 2017 to December 31, 2022. Patent base data downloaded from the State Intellectual Property Office of China. Detailed comparisons are made based on China's current National Statistical Classification of the Sports Industry (2019). China's sports industry is divided into 3 layers. The first level has 11 major categories, the second level has 37 medium categories, and the third level has 71 minor categories. Remove patents that do not conform to the Classification. As the number of patents in Hong Kong, Macao and Taiwan included in the

"Patent Search and Analysis System" is relatively small, it cannot correctly reflect the innovation capacity of these regions. Therefore, the study area is the 31 administrative regions in China mainland in this paper, excluding the above three regions.

## 2.2 Research methods

There weren't ethical issues involved in the research. There weren't relevant experiments.

**2.2.1 Natural breakpoint classification method.** The natural breakpoint classification method, also known as Jenks optimization method, divides the data into different classes and makes the results satisfy the best classification principle through an iterative process, i.e., the variance between each class is minimized, while the variance between classes is required to be maximized. In a word, the Jenks optimization method yields the smallest within-group variance and the largest between-group variance. This method is a good way to classify areas with similar properties and scores into the same level, and it is objective. Picture creation using mapping software GeoDa and ArcGIS.

**2.2.2 Exploratory Spatial Data Analysis (ESDA).** Exploratory spatial data analysis process refers to the use of various functions to gain an initial understanding of a spatial dataset [14]. ESDA is a collection of spatial data analysis methods and techniques to mine the spatial distribution characteristics of things by describing the spatial dependence and spatial heterogeneity of data [15]. The definition of spatial weight matrix is used to explain the spatial relationship between regions, and then extract the spatial connection and evolution pattern of complex socio-economic phenomena from them. ESDA suggests the mechanism of spatial interaction between phenomena, the core of which is the quantification of spatial structure and spatial location, spatial autocorrelation, involving the construction of spatial weight matrix, the measurement of global spatial autocorrelation, local spatial autocorrelation and the identification of spatial association [16]. In this paper, we use ArcGIS 10.8 and GeoDa spatial analysis software to realize relevant spatial data analysis.

*(1) Spatial weight matrix.* The spatial weight matrix is a way to quantify the relative spatial location and spatial interaction effects between spatial unit i and spatial unit j on a geographic space. It can reflect whether spatial cell i is adjacent to spatial cell j in terms of spatial structure. Given the large regional area differences among Chinese provinces, a spatial weight matrix based on geographical proximity is used.

In order to eliminate the effects of the geographical isolation of Hainan Island, Hainan Province was set up as a neighbor to Guangdong Province. Also, in order to eliminate the phenomenon of large variation in spatial effects due to too large values of attributes in a spatial cell, it is necessary to standardize the spatial weights using row normalization.

*(2) Global spatial autocorrelation.* The Global spatial autocorrelation reflects the spatial correlation of whole that mainland innovation capacity. The global spatial autocorrelation statistics such as Moran's I, Getis-Ord General G, and Global Geary's C were mainly estimated to analyze the spatial association and the degree of spatial variation of regional aggregates. In this paper, the most widely used Moran's I's are used as the statistic for testing global spatial correlation. Moran's I is calculated as follows:

$$I = \frac{n\sum_{i=1}^{n}\sum_{j=1}^{n}w_{ij}(x_i - \bar{x})(x_j - \bar{x})}{(\sum_{i=1}^{n}\sum_{j=1}^{n}w_{i,j})\sum_{i=1}^{n}(x_i - \bar{x})^2}$$

*(3) Local spatial autocorrelation.* Local autocorrelation can find the contribution of each spatial per-cell observation. Common statistics for testing local spatial autocorrelation are local Moran's I, LISA clustering map, Moran scatter plot, local Getis-Ord Gi*, etc. To better explore the evolution of China's sports industry innovation capabilities within each study region. In this paper, we use LISA clustering maps to detect the degree of spatial aggregation of observations of spatial units at the local level and analyze the association patterns of different regions.

**2.2.3 Spatio-temporal correlation analysis.** One of the typical uses of the bivariate Moran's I test is in the context of spatio-temporal analysis, where the bivariate Moran's I test introduces a unique perspective of spatio-temporal analysis. The only difference is that the spatial lag term of the bivariate Moran's I will no longer be its own spatial lag term but will be replaced by the spatial lag term of other variables. In the parameter setting of the test process, the variation of different time-invariant parameters leads to different analytical perspectives. A novel perspective of Time-space analysis can be provided in the empirical analysis of both outward spillover and inward spillover.

## 2.3 Research and research methodological limitations

1. Due to the small number of patents from Hong Kong, Macao and Taiwan included in the State Intellectual Property Office of China, the innovation capacity of these regions cannot be correctly reflected. Therefore, the research area of this paper is the 31 administrative regions in mainland China, excluding the three regions mentioned above, which results in the study not being able to completely reflect the innovation capacity of the sports industry in China as a whole.

2. EDSA is strongly influenced by geospatial location and the study areas need to be adjacent to each other. Hainan Island in China is geographically "isolated", but is the closest to Guangdong Province. In order to eliminate the effects of this phenomenon, Hainan Province and Guangdong Province were set to be adjacent to each other.

## 3. The spatial and temporal evolution process of innovation capacity of sports industry3.1 The development process of innovation capacity of sports industry

After obtaining the original basic data, screening, de-weighting, testing and other processes with obtaining the final data. Compile the changes in the number of patents of each province every year. There is also a clear variation in the number of patents per province. as shown in Fig 1.

As can be seen in Fig 1, the number of patents in the field of sports is increasing. The number of patents within the field of sports grew more moderately in the two years 2017 and 2018 in each province. The number of patents in the field of sports in each province grows rapidly from 2019 until it peaks in 2021, the same year that the annual growth in the number of patents in the field of sports in China reaches its maximum. China's entire sports industry and its ability to innovate have been greatly impacted by COVID-19. As a result, the number of patents declines in 2022. The growth rate of the number of patents within the field of sports in each province began to level off again after 2021, indicating the different levels of development of the innovation capacity of China's sports industry at different stages, but showing an overall trend of continuous increase.

From the local perspective, there is little difference in the increase in the number of patents within the field of sports in most provinces. There is no excessive gap between each other in

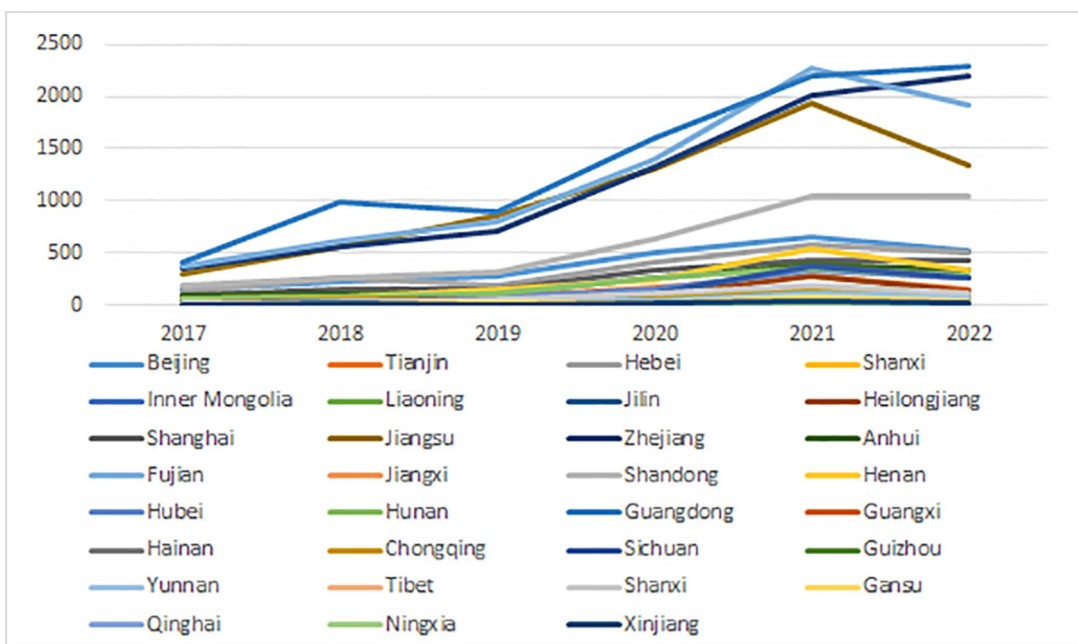

**Fig 1. Changes in the number of sports patents in each province from 2017 to 2022.**

terms of quantity, indicating that the gap in innovation capacity between most provinces is not large. Fujian, Zhejiang, Guangdong and Jiangsu provinces were in the lead in the number of patents in the field of sports from 2017. From 2018 onwards, it rapidly pulled away from the other provinces and was always in a distant position. The number of patents in the field of sports in Shandong also increased substantially in 2021 and began to catch up with the four provinces in front of it. but there is still a certain gap, which reflects that the innovation ability of Fujian, Zhejiang, Guangdong and Jiangsu provinces is always at a high level, while the innovation ability of Shandong's sports industry is not as good as the above four provinces. but it is ahead of the other provinces. The lines in the graph representing the provinces of Inner Mongolia, Hainan, Tibet, Qinghai, Xinjiang and Ningxia are almost a straight line, indicating the weak growth of the number of patents in the field of sports, reflecting the lack of innovation capacity of these provinces.

China's patent law divides the types of patents into invention patents, design patents and utility model patents. Compare with 2017, the number of patents of the three types shows a rising trend. It shows that the overall situation of the innovation ability of the sports industry is better, with the most significant growth in the number of utility model patents and relatively weak growth in the number of invention patents. The proportion of invention patents in the overall patents during the period was 25.1%, 17.8%, 17.1%, 12.2%, 10%, 8.2% and 12.1% respectively, and the proportion of invention patents showed a clear downward trend. This reflects the problem of low quality of innovation capacity of sports industry, and the "quality" cannot match the "quantity" in the process of improving innovation capacity of sports industry. as shown in Fig 2.

## 3.2 Characteristics of the evolution of the capacity pattern of the sports industry between provinces

To show more intuitively the characteristics of the evolution of the spatial pattern of the innovation capacity of China's sports industry. The number of patents in the field of sports in the 31

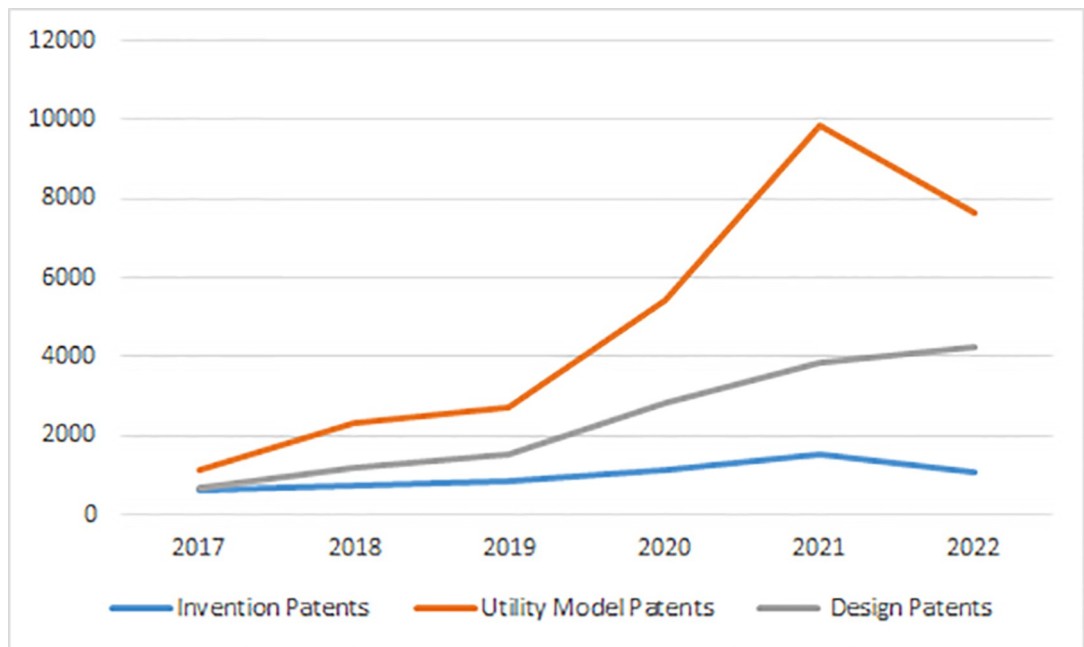

**Fig 2. Folding chart of the temporal distribution of the three types of patents in the sports industry.**

provinces of China is used as the basis for classification. The natural breakpoint classification was used to classify the innovation capability of China's sports industry at each stage into four hierarchical types: low level, medium level, higher level and high level. The redder the color shown on the map, the higher the rank of the province's innovation capacity, and the lighter the color shown on the map, the lower the rank of the province's innovation capacity. as shown in Fig 3.

According to Fig 3, It can be found that the number of provinces within the four innovation capacity class types fluctuates less as time advances, although the class types in which the provinces are located have changed. From the level of sports industry innovation capacity, the spatial distribution of sports industry innovation capacity varies significantly, and the provinces with higher and high-level level types of sports industry innovation capacity are mainly concentrated in the eastern coastal region, provinces with low level rating type of sports industry innovation capacity are mainly concentrated in the central and western inland areas, showing the dependence on the level of sports industry development. The innovation capacity of the sports industry as a whole has formed with the Pearl River Delta, Yangtze River Delta, Beijing-Tianjin-Hebei and the economic circle on the west coast of the Taiwan Strait, which are regions with a high level of sports industry development, as the leading core, Taking the major provinces in these economic circles as the starting point of the spatial pattern, the regions with high innovation capacity levels gradually extend continuously from the eastern coastal areas to the central and western inland regions. Due to innovation in the sports industry is inextricably linked to the economic base. China's coastal areas have a high level of economic development, rich material means of living, rapid circulation of resource elements, and more investment in sports science and technology. Therefore, the East China Sea coastal area has a better level of innovation capacity in the sports industry. it explains why the provinces with a low level of innovation capacity in the sports industry are mainly concentrated in the central and western inland regions.

In terms of different evolutionary stages. In each stage of the sports industry innovation capacity in the higher, high level grade type of provinces are always a few, mainly Guangdong, Fujian, Zhejiang, Jiangsu, Shandong provinces, and Guangdong is always in a high level state.

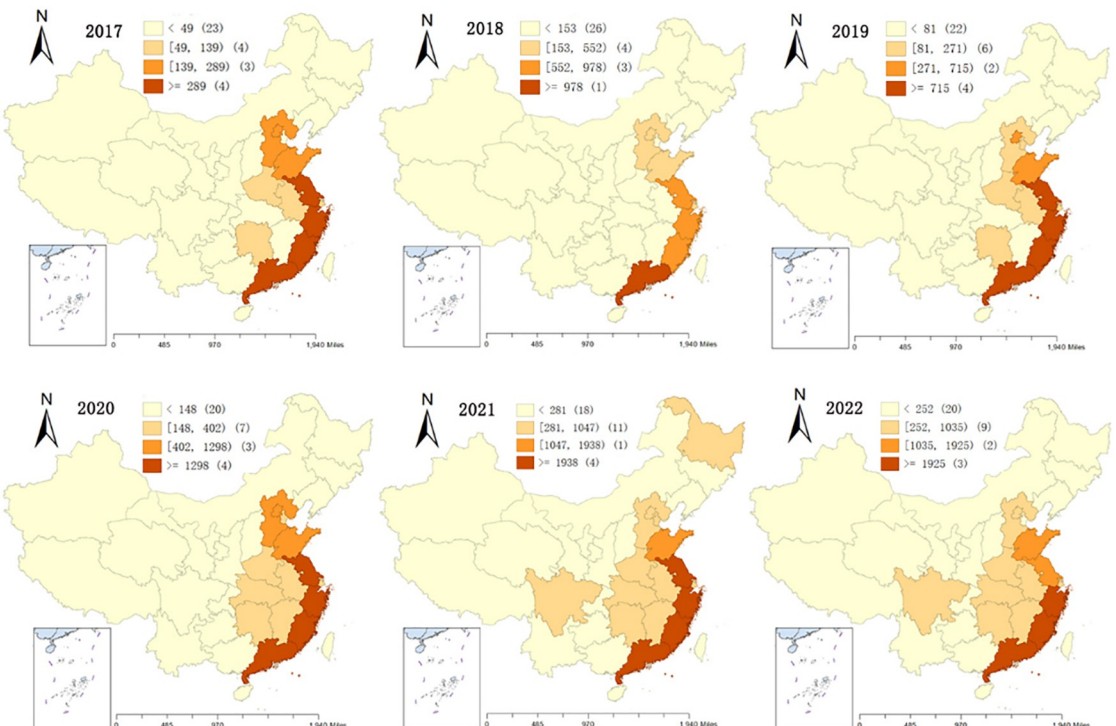

**Fig 3. Evolution process of spatial pattern of sports industry innovation capability from 2017 to 2022.**

Although there is an increase in the number of patents in the central and western inland provinces, but most of them are always in the low level rank type. The innovation capacity of the sports industry has basically formed a spatial pattern with the eastern coastal region as the core of the agglomeration.

### 3.3 Spatial correlation characteristics of the innovation capability of sports industry

**3.3.1 Global spatial autocorrelation analysis of the innovation capability of sports industry.** Fig 3 can only reflect the evolution of the spatial pattern of the innovation capacity of the sports industry, In order to further verify whether there is spatial clustering and spatial correlation characteristics of the innovation capability of sports industry in space, the spatial correlation of the innovation capability of sports industry in the study area is achieved through the analysis and test of global autocorrelation as a whole. Moran's I correlation test for global spatial autocorrelation of innovation capability of sports industry at each stage was conducted separately using spatial data analysis software, as shown in Table 1.

**Table 1. 2017–2022 global Moran's I test table.**

| Year | Moran's I | E(I) | Z(I) | P |
|---|---|---|---|---|
| 2017 | 0.337 | -0.033 | 3.229 | 0.005 |
| 2018 | 0.225 | -0.033 | 2.384 | 0.024 |
| 2019 | 0.287 | -0.033 | 2.829 | 0.011 |
| 2020 | 0.327 | -0.033 | 3.167 | 0.006 |
| 2021 | 0.330 | -0.033 | 3.177 | 0.006 |
| 2022 | 0.320 | -0.033 | 3.144 | 0.007 |

As seen in Table 1, the values of the global Moran's I index of innovation capacity of the sports industry at each stage are greater than 0 and pass the significance test. It shows that there is a spatial clustering of innovation capabilities in China's sports industry. The value of global Moran's I in 2017 is greater than the value of global Moran's I in subsequent years, indicating that the innovation capacity of China's sports industry is most spatially clustered in 2017. The value of the global Moran's I is consistently increases from 2018 to 2021, indicating that the concentration of innovation capabilities in China's sports industry has been strengthened during this time period. The value of global Moran's I in 2022 is slightly smaller than the value of global Moran's I in 2021 and 2020, which indicates that the degree of clustering of innovation capacity in China's sports industry in 2022 is smaller than that in the first two years, but the difference is not large. The development trend of the spatial agglomeration degree of China's sports industry innovation capability is reflected as follows: first decreasing, then increasing and finally leveling off.

In general, the global Moran's I values show a tendency to increase year by year. This indicates that the agglomeration of innovation capabilities in China's sports industry is in a strengthened state and the spatial correlation under the spillover effect is gradually increasing

**3.3.2 Local spatial autocorrelation analysis of innovation capacity of sports industry.** The global Moran's I measures the degree of spatial autocorrelation of sports industry innovation capacity within the study region as a whole, but it does not further reveal the spatial autocorrelation of sports industry innovation capacity among specific provinces, which cannot reflect the exact characteristics and patterns of spatial clustering of sports industry innovation capacity among regions. Therefore, the LISA diagram is used to further identify and analyze the correlation patterns of different localities, as shown in Fig 4.

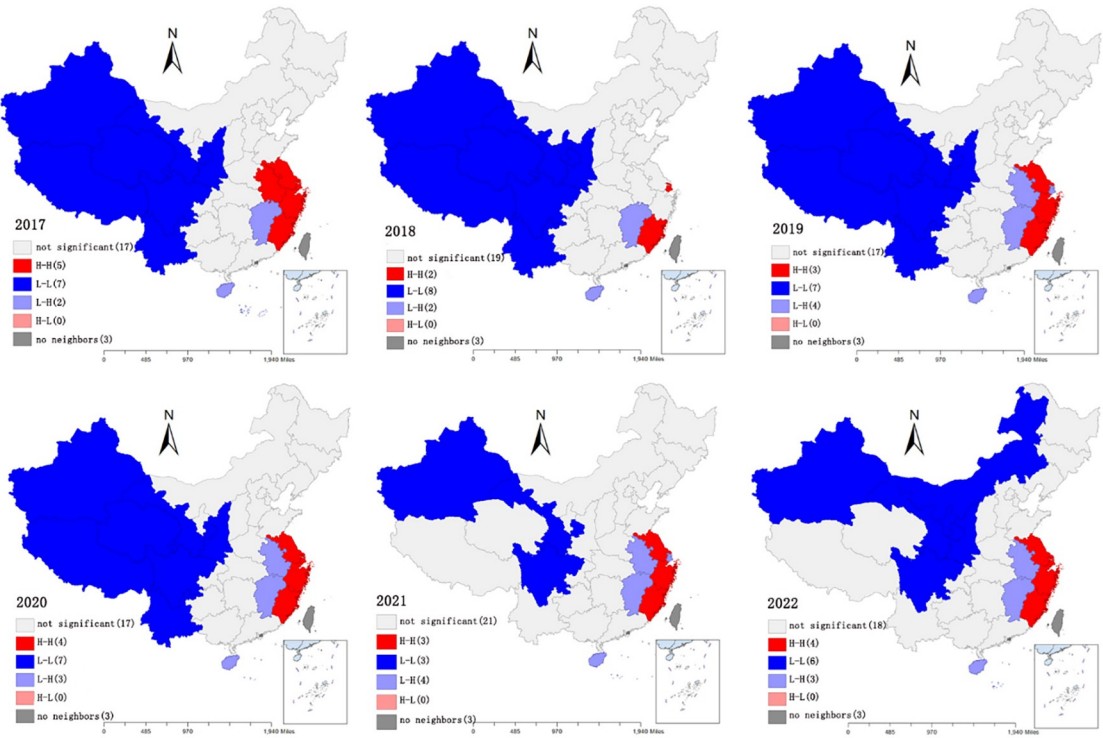

**Fig 4. 2017–2022 LISA chart of China's sports industry innovation capacity.**

1. *High-high cluster area (efficient type)*. The provinces within the high—high agglomeration of sports industry innovation capacity from 2017 almost all are Shanghai, Jiangsu, Zhejiang and Fujian. These provinces have a higher capacity for innovation in the sports industry than their neighbors, creating a high-value agglomeration of sports industry capabilities. Although Guangdong itself is in the type of high level of innovation capacity of sports industry and is also a neighbor of Fujian Province, it is not in this agglomeration and belongs to the insignificant area because the spatial weights are row normalized and there are more neighboring provinces with low level of innovation capacity.

2. *Low-high agglomeration area (polarization type)*. The provinces in the low-high concentration area of sports industry innovation capacity are mainly Anhui, Jiangxi and Hainan. Anhui and Jiangxi are both neighboring provinces belonging to the high-high agglomeration area, and the gap between the innovation capacity of the provinces in the high-high agglomeration area is too large, while the surrounding provinces are insignificant areas, i.e. no spatial agglomeration is produced, thus forming a low-high agglomeration area for the innovation capacity of the sports industry. Hainan Province has only one neighbor, Guangdong, and its own sports industry innovation capacity is low level type while Guangdong's sports industry innovation capacity is always in high level type which is also in low-high agglomeration by this influence. Shanghai is surrounded completely by Jiangsu and Zhejiang in which is high levels of sports industry innovation capacity, and its own sports industry innovation capacity is lower than these two provinces. Therefore presenting a high-low agglomeration.

3. *Low-low agglomeration area (low efficiency type)*. The provinces within the high-low concentration area of the innovation capacity of the sports industry are all inland provinces from the northwest from 2017 to 2020, and three provinces, Yunnan, Tibet, and Qinghai, are detached from the region from 2021, and Inner Mongolia becomes into the low-low concentration area in 2022. The innovation capacity of the sports industry in these provinces is at a low level, while the differences with the neighboring provinces are small, thus forming a low-value agglomeration of sports industry capacity.

4. *Insignificant area (random distribution)*. The provinces in this part of the region fail to form spatial agglomeration or spatial dispersion with their neighboring provinces in local space, i.e., Moran's I = 0 in the test of local spatial autocorrelation, which is randomly distributed in space.

## 4. Analysis of the development trend of innovation ability of sports industry

### 4.1 Interactive phenomenon of innovation capability of sports industry

On the one hand, the natural breakpoint classification map has presented a continuous extension of China's sports industry innovation capacity from the eastern coastal region to the central and western inland regions. Meanwhile, according to the above study, it is proved that there is a significant spatially positive and increasing correlation between the innovation capacity of Chinese sports industry. In the process of data collation, it is found that the phenomenon of mutual ownership of patents among provinces is becoming more and more obvious, mainly manifested in the increasing number of provinces with common patents from 2017 to 2022, and the number of patents shared among different provinces is increasing as shown in Table 2.

The above research shows that the innovation capacity of Chinese sports industry not only has a clustering effect in the spatial scope but also has mobility in the evolution of spatial

**Table 2. Interaction table of the number of patents within the sports neighborhood between provinces, 2017–2022.**

| 2017 | 2018 | 2019 | 2020 | 2021 | 2022 |
|---|---|---|---|---|---|
| Beijing\Jiangsu | Beijing\Tianjin | Beijing\Shandong | Anhui\Fujian | Beijing\Heilongjiang | Beijing\Guangdong |
| Beijing\Shanghai | Guangdong\Jiangsu | Guangdong\Beijing | Beijing\Guizhou | Beijing\Jilin | Beijing\Guizhou |
| Beijing\Tianjin | Guangdong\Shandong | Guangdong\Hunan | Beijing\Jilin | Beijing\Jiangsu | Beijing\Heilongjiang |
| Guangdong\Beijing\Hebei | Hubei\Beijing | Guangdong\Jiangxi | Beijing\Jiangsu | Beijing\Liaoning | Beijing\Sichuan |
| Guangdong\Guangxi | Shandong\Shanxi | Guangdong\Shandong | Beijing\Tianjin | Beijing\Shanxi | Beijing\Zhejiang |
| Hubei\Beijing | Shanghai\Beijing | Guangdong\Zhejiang | Fujian\Shanxi | Fujian\Anhui | Beijing\Shanghai |
|  | Shanghai\Zhejiang | Hubei\Beijing | Guangdong\Beijing | Fujian\Heilongjiang | Fujian\Guangdong |
|  |  | Sichuan\Henan | Guangdong\Hubei | Fujian\Zhejiang | Fujian\Heilongjiang |
|  |  | Zhejiang\Beijing | Guangdong\Hunan | Guangdong\Anhui | Fujian\Hebei |
|  |  | Zhejiang\Shandong | Guangdong\Shandong | Guangdong\Beijing | Fujian\Jiangsu |
|  |  |  | Heilongjiang\Anhui | Guangdong\Fujian | Fujian\Shandong |
|  |  |  | Heilongjiang\Beijing\Shandong | Guangdong\Hunan | Guangdong\Hunan |
|  |  |  | Hunan\Guangdong | Guangdong\Shandong | Guangdong\Jiangsu |
|  |  |  | Shandong\Guangxi | Hainan\ Guangdong | Guangdong\Shandong |
|  |  |  | Shanghai\Guangdong | Hebei\Hubei | Guangdong\Shanghai |
|  |  |  | Shanghai\Jiangsu | Hubei\Shanxi | Guangdong\Zhejiang |
|  |  |  | Shanghai\Shandong | Shandong\Beijing | Hubei\Shanxi |
|  |  |  | Sichuan\Henan | Shandong\Shanghai | Shandong\Hebei |
|  |  |  | Zhejiang\Fujian | Shanxi\Zhejiang | Shanghai\Shandong |
|  |  |  | Zhejiang\Jiangsu | Shanghai\Jiangsu | Zhejiang\Sichuan |
|  |  |  |  | Shanghai\Zhejiang |  |
|  |  |  |  | Tianjin\Beijing |  |
|  |  |  |  | Zhejiang\Guangdong |  |
|  |  |  |  | Zhejiang\Jiangsu |  |
|  |  |  |  | Zhejiang\Sichuan |  |
| Number of jointly held patents |  |  |  |  |  |
| 10 | 17 | 15 | 47 | 60 | 45 |

distribution pattern, and the spatial structure has changed accordingly, and the innovation capacity of sports industry among spatial units is not a state of mutual isolation.

On the other hand, theories related to geographical economics have revealed that agglomeration and diffusion mechanisms are the basic dynamic mechanisms for the evolution of regional spatial structure, and the agglomeration effect makes regional innovation capacity develop from an isolated, dispersed homogeneous and disorderly state to a low-level orderly state with local uneven development, and the diffusion effect makes regional innovation capacity develop into a relatively balanced high-level orderly state [17]. The process of "agglomeration—diffusion—reagglomeration—rediffusion" continues to develop spatially like a wave, causing the regional spatial structure to evolve [18].

Therefore, given the actual situation of the spatio-temporal evolution process of the innovation capability of Chinese sports industry is consistent with the relevant theories. In this paper, the following hypothesis is proposed: there is a spatial spillover effect on the innovation capability of Chinese sports industry.

## 4.2 Validation and trend analysis of Innovation capability spillover effect

**4.2.1 Variable setting and interpretation of spatio-temporal correlation analysis.** To further confirm the existence of spatial spillover effects on the innovation capability of China's

sports industry, a spatio-temporal correlation analysis was introduced to verify this using the novel analytical perspective of the bivariate Moran's I test. Spatial overflow manifests itself as either inward or outward overflow, so the variables need to be set accordingly when performing the bivariate Moran's I test variables.

1. t represents time, and the number of patents in the field of sports in year (t) is taken as the first variable (X), and the number of patents in the field of sports in year (t-1) is taken as the second variable (Y). Described under this setting is the spatial interaction of the number of patents within the sports domain in one province in year t by the average annual weighted number of patents within the sports domain in neighboring provinces (t-1). Although space is interactively influenced, the time factor fixes that the influence of space-time is uni-directional, and it can only be that the innovation ability in (t-1) year influences the innovation ability in (t) year, but not the innovation ability in (t) year influences the innovation ability in (t-1) year. If Moran's I > 0 can be seen as an inward spillover of the innovation capacity of the sports industry, i.e., spillover from the past neighborhood region to the present core region.

2. The number of patents in the field of sports in year (t-1) is taken as the first variable (X), and the number of patents in the field of sports in year (t) is taken as the second variable (Y). From the unidirectional influence relationship in time, it clear that setting describes the spatio-temporal interaction of the number of patents in the field of sports in one province in (t-1) years on the weighted average number of patents in the field of sports in its neighboring provinces in the future (t) years. If Moran's I > 0 can be seen as an outward spillover of the innovation capacity of the sports industry, i.e., spillover from the core region in the past to the neighboring region in the present.

**4.2.2 Spatio-temporal correlation analysis test results and trend analysis.** The test of inward spillover of innovation capability of Chinese sports industry is carried out according to setting (1), and the details are shown in Table 3.

Table 3 shows that the values of the bivariate Moran's index are greater than 0 and pass the significance test in the test of inward spillover of innovation capacity of Chinese sports industry from 2018 to 2022. Evidence of inward spatial spillover of innovation capabilities in China's sports industry. Although the degree of inward spillover of the innovation capacity of China's sports industry decreased compared with 2017, the inward spillover of the innovation capacity of China's sports industry has been strengthening from 2018 to 2022. The direction of innovation capacity spillover from the neighboring regions in year (t-1) to the core region in year (t) in the sports industry.

The above tests prove that the innovation capacity of the sports industry overflows from the neighboring regions in the past to the core regions in the present every year, and the core

**Table 3. Inward spillover test of China's sports industry innovation capability.**

| t | Moran's I | E(I) | Z(I) | P |
|---|---|---|---|---|
| 2017 | | | | |
| 2018 | 0.282 | -0.033 | 2.822 | 0.011 |
| 2019 | 0.257 | -0.033 | 2.608 | 0.016 |
| 2020 | 0.307 | -0.033 | 2.955 | 0.008 |
| 2021 | 0.324 | -0.033 | 3.139 | 0.006 |
| 2022 | 0.328 | -0.033 | 3.194 | 0.007 |

Table 4. Outward spillover test of China's sports industry innovation capability.

| t-1 | Moran's I | E(I) | Z(I) | P |
|---|---|---|---|---|
| 2017 | 0.276 | -0.033 | 2.768 | 0.012 |
| 2018 | 0.260 | -0.033 | 2.637 | 0.016 |
| 2019 | 0.310 | -0.033 | 3.025 | 0.008 |
| 2020 | 0.333 | -0.033 | 3.218 | 0.005 |
| 2021 | 0.332 | -0.033 | 3.233 | 0.006 |
| 2022 | | | | |

regions with high value concentration are mainly Zhejiang, Fujian and Jiangsu provinces, which become the main regions for the overflow. This explains the reason for the rapid increase in the number of patents in the field of sports in these three provinces. Therefore, Zhejiang, Fujian and Jiangsu provinces are in a high level of innovation capacity in the sports industry. According to the test results of inward spillover of sports industry innovation capacity, it can be predicted that in 2023 the innovation capacity of China's sports industry will continue to spill over from the neighboring provinces in the 2022 agglomeration area to the provinces in the 2023 agglomeration core, and more sports industry innovation capacity will spill over into Zhejiang, Fujian and Jiangsu provinces.

The test of outward spillover of innovation capability of Chinese sports industry is conducted according to setting (2), and the details are shown in Table 4.

Similarly, Table 4 shows that the values of the bivariate Moran's index are greater than 0 and pass the significance test in the test of outward spillover of the innovation capacity of Chinese sports industry from 2018 to 2022, which proves that there is outward spillover of the innovation capacity of Chinese sports industry in space. The trend of outward spillover of innovation capacity in China's sports industry continues to rise from 2018 to 2020 and starts to level off, with an overall upward trend. The direction of innovation capacity spillover from the core region in year (t-1) to the neighboring region in year (t) for the sports industry.

The above tests prove that the innovation capacity of the sports industry spills over from the core regions of the past to the neighboring regions of the present every year. Due to the difference in the level of innovation capacity, the value of innovation capacity spillover of sports industry in high-value agglomeration core areas is much higher than that in low-value agglomeration core areas, and due to the geographical proximity, the innovation capacity of sports industry is more likely to spillover to neighboring provinces in high-value agglomeration core areas. This explanation that the provinces of Guangdong, Hubei, Hebei, Hunan, Sichuan, and Beijing are still in higher and high level rank of sports industry innovation capacity, although they are not in the spatial agglomeration area. According to the test results of outward spillover of sports industry innovation capacity, it can be predicted that in 2023, the innovation capacity of China's sports industry will continue to spill over from the provinces in the core of the concentration in 2022 to the neighboring provinces in the core of the concentration in 2023, and the innovation capacity of China's sports industry will spill over more to Guangdong, Hubei, Hebei, Hunan, Sichuan, and Beijing.

## 5. Conclusions and recommendations

### 5.1 Conclusion

**5.1.1 Wide regional gaps in spatial innovation capability of China's sports industry.**
The innovation ability of sports industry depends on the level of sports industry development. The innovation capacity of China's sports industry in general shows a growing trend, while the

overall sports industry innovation capacity is rising, the gap in the innovation capacity of the sports industry between regions is also increasing, with more than half of the provinces' sports industry innovation capacity still at a low level. The spatial structure shows a clear pattern of Strong in the east and weak in the west, especially reflected in the extremely strong southeast coastal areas and extremely weak northwest inland areas, and the gap between the area is too large.

**5.1.2 There is an obvious spatial agglomeration of innovation capability in China's sports industry.** The phenomenon of spatial clustering of innovation capacity in China's sports industry is gradually increasing. The innovation capacity of sports industry in high-high agglomeration regions has an efficient state, mainly distributed in Fujian, Zhejiang and Jiangsu provinces on the southeast coast. The innovation capacity of sports industry in high-low agglomeration and low-high agglomeration regions has a polarized state, mainly distributed in the periphery of high-value agglomeration provinces. The innovation capacity of sports industry in low-low agglomeration areas is relatively insufficient and mainly distributed in the northwest inland area.

**5.1.3 There is a spatial spillover effect on the innovation capability of China's sports industry.** There is a spatial spillover effect on the innovation capacity of China's sports industry, with inward spillover continuing to strengthen mainly into Zhejiang, Fujian and Jiangsu provinces. The outward spillover is mainly to Guangdong, Hubei, Hebei, Hunan, Sichuan and Beijing, which are the provinces and cities, especially those adjacent to the core area. The innovation capacity of sports industry in these provinces is growing faster.

## 5.2 Recommendations

**5.2.1 Promote the development of sports industry in the northwest region.** Do a good job of government policy guidance, talent training, investment, risk protection and other aspects of work to promote the development of the characteristics of the northwest regional sports industry, clustering, the construction a number of cultural enterprises with a high degree of concentration, industrial core competitiveness of the sports industry clusters. Based on the agglomeration of sports industry, according to the actual situation of the development of regional sports industry, we enhance the ability to accurately invest in resources related to sports industry and promote the collaborative development of industry, academia and research, in order to increase the output of regional sports industry innovation results, so as to enhance the innovation capacity of regional sports industry.

**5.2.2 Play the advantages of regional characteristics of sports industry.** The phenomenon of local agglomeration of innovation capacity of sports industry is obvious, and it is not only necessary to make full use of the agglomeration and diffusion mechanism with the advantage of location development, but also to promote the diversification of sports industry by combining the advantages of resource endowment and functional positioning of regional development. Realize the division of labor between provinces in the spatial agglomeration area and between sports industries which form a sports industry chain and innovation chain with regional characteristics. To make the sports industry innovation capacity accumulate, accelerate the continuous evolution of spatial structure, further enhance the innovation capacity of sports industry in high—high agglomeration regions. Promote the development of high—low agglomeration, low—high agglomeration, low—low agglomeration regional sports industry innovation capacity.

**5.2.3 Accelerate the integration of neighboring regions to build the process.** The spatial spillover effect of sports industry innovation capacity continues to strengthen, but the construction of China's unified large market is not complete, and various barriers still exist

between regions, which gradually break their respective barriers through the construction of neighboring regional integration and promote the free flow of sports industry innovation factors within the region. According to the spatial layout and spatial spillover effect of the innovation capacity of sports industry, gradually advancing from the eastern coastal areas to the western inland areas. Then, To lead the development of sports industry innovation capacity in regions with relatively low level of sports industry innovation capacity.

## Author Contributions

**Data curation:** Jiang Chen.

**Formal analysis:** Jiang Chen, ChuanMing Cai.

**Methodology:** Jiang Chen.

**Project administration:** ChuanMing Cai.

**Resources:** Jiang Chen, TingTing Zhang.

**Software:** Jiang Chen.

**Supervision:** TingTing Zhang.

**Visualization:** ChuanMing Cai.

**Writing – original draft:** Jiang Chen.

**Writing – review & editing:** Jiang Chen.

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
