## [Decision Letter · Decision Letter 0]

17 Jul 2023

PONE-D-23-09406Analysis of the spatial and temporal evolution process and development trend of innovation capability of Chinese sports industryPLOS ONE

Dear Dr. Cai,

Thank you for submitting your manuscript to PLOS ONE. After careful consideration, we feel that it has merit but does not fully meet PLOS ONE’s publication criteria as it currently stands. Therefore, we invite you to submit a revised version of the manuscript that addresses the points raised during the review process.

We look forward to receiving your revised manuscript.

Kind regards,

Abdulkader Murad, Ph.D

Academic Editor

PLOS ONE

Journal Requirements:

4. We note that Figures 2 and 3 in your submission contain map/satellite images which may be copyrighted. All PLOS content is published under the Creative Commons Attribution License (CC BY 4.0), which means that the manuscript, images, and Supporting Information files will be freely available online, and any third party is permitted to access, download, copy, distribute, and use these materials in any way, even commercially, with proper attribution. For these reasons, we cannot publish previously copyrighted maps or satellite images created using proprietary data, such as Google software (Google Maps, Street View, and Earth). For more information, see our copyright guidelines: http://journals.plos.org/plosone/s/licenses-and-copyright.

a. You may seek permission from the original copyright holder of Figures 2 and 3 to publish the content specifically under the CC BY 4.0 license.  

Additional Editor Comments (if provided):

Reviewer 1

he manuscript has the required information and is written in good English grammar, however more literature must be added. The paper has more information in economics then physical education and sports can can be considered in other journals,

Reviewer 2

1- Are there any examples of high-quality development of sport industry that has been applied and succussed in other countries? Or maybe in China but in other fields. Please include some evidenced examples.

2- Please provide details of inclusion and exclusion criterions of the patents data used in the study.

3- Please provide more details on the nature of the distribution of the data? for example, what kind of sports are included? Are there any particular sports that has noticeably more data size than other? If yes, would that effect the study results?

4- Why there is a noticeable drop of sports patents in all provinces in 2022 compared to 2021? Could that be related to the coronavirus pandemic?

5- “… higher and high-level level types of sports industry innovation capacity are mainly concentrated in the eastern coastal region...” are there any correlation between sports industry innovation capacity and financial level of the province? In other words, economy could be one of the main reasons for such distribution. Please elaborate.

6- Define acronyms in Table1, Table3, and Table4

7- Figure3: use English in Figur3 labels.

8- What are the limitations of the current study?

Reviewers' comments:

Reviewer's Responses to Questions

**Comments to the Author**

1. Is the manuscript technically sound, and do the data support the conclusions?

Reviewer #1: Yes

Reviewer #2: Yes

2. Has the statistical analysis been performed appropriately and rigorously? 

Reviewer #1: Yes

Reviewer #2: Yes

3. Have the authors made all data underlying the findings in their manuscript fully available?

Reviewer #1: Yes

Reviewer #2: No

4. Is the manuscript presented in an intelligible fashion and written in standard English?

Reviewer #1: Yes

Reviewer #2: Yes

5. Review Comments to the Author

Reviewer #1: The manuscript has the required information and is written in good English grammar, however more literature and be added. The paper has more information n economics then physical education and sports can can be considered in other journals

Reviewer #2: 1- Are there any examples of high-quality development of sport industry that has been applied and succussed in other countries? Or maybe in China but in other fields. Please include some evidenced examples.

2- Please provide details of inclusion and exclusion criterions of the patents data used in the study.

3- Please provide more details on the nature of the distribution of the data? for example, what kind of sports are included? Are there any particular sports that has noticeably more data size than other? If yes, would that effect the study results?

4- Why there is a noticeable drop of sports patents in all provinces in 2022 compared to 2021? Could that be related to the coronavirus pandemic?

5- “… higher and high-level level types of sports industry innovation capacity are mainly concentrated in the eastern coastal region...” are there any correlation between sports industry innovation capacity and financial level of the province? In other words, economy could be one of the main reasons for such distribution. Please elaborate.

6- Define acronyms in Table1, Table3, and Table4

7- Figure3: use English in Figur3 labels.

8- What are the limitations of the current study?

6. PLOS authors have the option to publish the peer review history of their article (what does this mean?). If published, this will include your full peer review and any attached files.

Reviewer #1: **Yes: **Dr Mohammed Feroz Ali

Reviewer #2: **Yes: **Zainab Altai

---

## [Author Response · Author response to Decision Letter 0]

22 Jul 2023

I will respond to each of them here. The details are as follows.

1. Please ensure that your manuscript meets PLOS ONE's style requirements

Response: My manuscript would meet PLOS ONE's style requirements

2. In your Data Availability statement, you have not specified where the minimal data set underlying the results described in your manuscript can be found.

Response: The minimal data set in the manuscript were obtained from the database of the China Intellectual Property Office (CIPO).

The relevant data are open data, and the use of them is not restricted in any way. Anyone who registers as a user will be able to collect data on website.

url: https://pss-system.cponline.cnipa.gov.cn/conventionalSearch

3. Please include your full ethics statement in the ‘Methods’ section of your manuscript file.

Response: I include my full ethics statement in the ‘Methods’ section of your manuscript file. Claim: Absolutely, there weren’t ethical issues involved in the research. There weren’t relevant experiments.

4. We note that Figures 2 and 3 in your submission contain map/satellite images which may be copyrighted. All PLOS content is published under the Creative Commons Attribution License (CC BY 4.0), which means that the manuscript, images, and Supporting Information files will be freely available online, and any third party is permitted to access, download, copy, distribute, and use these materials in any way, even commercially, with proper attribution. For these reasons, we cannot publish previously copyrighted maps or satellite images created using proprietary data, such as Google software (Google Maps, Street View, and Earth). For more information, see our copyright guidelines: http://journals.plos.org/plosone/s/licenses-and-copyright.

Response: Figures 2 and 3 do not involve any copyright issues. Images were created by the author using publicly available data. Picture creation using mapping software GeoDa and ArcDIS.

5. Please review your reference list to ensure that it is complete and correct.

Response: I have checked. I have revised the wrong reference file format as required.

6. He manuscript has the required information and is written in good English grammar, however more literature must be added.

Response: I added some literature. However, the main feature of the research methodology is that it is very objective. It is a quantitative research method. As article is based on data analysis, it is difficult to add to many additions.

7. Are there any examples of high-quality development of sport industry that has been applied and succussed in other countries? Or maybe in China but in other fields. Please include some evidenced examples.

Response: DJI is a very successful case. DJI pays great attention to innovation capability. DJI has accumulated more than 4,600 patent applications. In terms of international patent applications, DJI ranks 29th in the world. DJI accounted for over 80% of the global market share in 2020 and over 70% domestically, ranking first among global civil drone companies. In the four years from 2013 to 2017, DJI's revenue has grown from 820 million all the way to 17.57 billion, and moreover, in 2020, it achieved a revenue of 26 billion.

8. Please provide details of inclusion and exclusion criterions of the patents data used in the study.

Response: Patent base data downloaded from the State Intellectual Property Office of China. Detailed comparisons are made based on China's current National Statistical Classification of the Sports Industry (2019). China's sports industry is divided into 3 layers. The first level has 11 major categories, the second level has 37 medium categories, and the third level has 71 minor categories. Remove patents that do not conform to the Classification. 

9. Please provide more details on the nature of the distribution of the data? for example, what kind of sports are included? Are there any particularly sports that has noticeably more data size than other? If yes, would that effect the study results?

Response: According to the research content as well as the research framework, I have already made further additions to the relevant information in the manuscript. According to the classification of patents in China, different types of patents are further discussed and analysis.

10.Why there is a noticeable drop of sports patents in all provinces in 2022 compared to 2021? Could that be related to the coronavirus pandemic?

Response: Yes. China's entire sports industry and its ability to innovate have been greatly impacted by COVID-19. As a result, the number of patents declines in 2022.

11. “… higher and high-level level types of sports industry innovation capacity are mainly concentrated in the eastern coastal region...” are there any correlation between sports industry innovation capacity and financial level of the province? In other words, economy could be one of the main reasons for such distribution. Please elaborate.

Response: According to public data and studies in China, innovation in the sports industry is inextricably linked to the economic base. China's coastal areas have a high level of economic development, rich material means of living, rapid circulation of resource elements, and more investment in sports science and technology. Therefore, the East China Sea coastal area has a better level of innovation capacity in the sports industry. 

12. Define acronyms in Table1, Table3, and Table4

Response: patent 17 indicates that the patent is dated in 2017. patent 18 indicates that the patent is dated in 2018. And so on, indicating patents in 2019, 2020, 2021, and 2022, respectively. For the convenience of the reader, these have been revised to 2017, 2018, 2019, 2020, 2021, and 2022 in the revised submission. 

13. Figure3: use English in Figur3 labels.

Response: It's been amended in the revised manuscript.

14. What are the limitations of the current study?

Response: The limitations are mainly two: (1) Due to the small number of patents from Hong Kong, Macao and Taiwan included in the State Intellectual Property Office of China, the innovation capacity of these regions cannot be correctly reflected. Therefore, the research area of this paper is the 31 administrative regions in mainland China, excluding the three regions mentioned above, which results in the study not being able to completely reflect the innovation capacity of the sports industry in China as a whole.

(2) EDSA is strongly influenced by geospatial location and the study areas need to be adjacent to each other. Hainan Island in China is geographically "isolated", but is the closest to Guangdong Province. In order to eliminate the effects of this phenomenon, Hainan Province and Guangdong Province were set to be adjacent to each other.

Thank you for your careful reading. Please contact me if there are any deficiencies and I will make changes in a timely manner.

---

## [Editor Report · Decision Letter 1]

31 Jul 2023

Analysis of the spatial and temporal evolution process and development trend of innovation capability of Chinese sports industry

PONE-D-23-09406R1

Dear Dr. Cai,

We’re pleased to inform you that your manuscript has been judged scientifically suitable for publication and will be formally accepted for publication once it meets all outstanding technical requirements.

Kind regards,

Abdulkader Murad, Ph.D

Academic Editor

PLOS ONE
---

## [Editor Report · Acceptance letter]

2 Aug 2023

PONE-D-23-09406R1 

Analysis of the spatial and temporal evolution process and development trend of innovation capability of Chinese sports industry 

Dear Dr. Cai:

I'm pleased to inform you that your manuscript has been deemed suitable for publication in PLOS ONE. Congratulations! Your manuscript is now with our production department. 

Kind regards, 

on behalf of

Professor Abdulkader Murad 

Academic Editor

PLOS ONE